

# Exceptional Chern-Simons-Matter dualities

**Clay Córdova[1⋆], Po-Shen Hsin [2†] and Kantaro Ohmori [1‡]**

**1** School of Natural Sciences, Institute for Advanced Study, Princeton, NJ, USA
**2** Walter Burke Institute for Theoretical Physics,
California Institute of Technology, Pasadena, CA, USA

⋆ claycordova@ias.edu, † phsin@caltech.edu, ‡ kantaro@ias.edu,

## Abstract

We use conformal embeddings involving exceptional affine Kac-Moody algebras to derive new dualities of three-dimensional topological field theories. These generalize the familiar level-rank duality of Chern-Simons theories based on classical gauge groups to the setting of exceptional gauge groups. For instance, one duality sequence we discuss is $(E_N)_1 \longleftrightarrow SU(9-N)_{-1}$. Others such as $SO(3)_8 \longleftrightarrow PSU(3)_{-6}$, are dualities among theories with classical gauge groups that arise due to their embedding into an exceptional chiral algebra. We apply these equivalences between topological field theories to conjecture new boson-boson Chern-Simons-matter dualities. We also use them to determine candidate phase diagrams of time-reversal invariant $G_2$ gauge theory coupled to either an adjoint fermion, or two fundamental fermions.

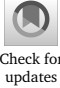
# 1 Introduction

In this paper we derive new dualities of three-dimensional Chern-Simons theories, and use them to propose new dualities of Chern-Simons-matter theories. Our results for TQFTs generalize the familiar level-rank dualities of Chern-Simons theories with classical gauge groups which are of the form:

$$SU(N)_K \longleftrightarrow U(K)_{-N,-N} , \quad U(N)_{K,K\pm N} \longleftrightarrow U(K)_{-N,-N\mp K} , \qquad Sp(N)_K \longleftrightarrow Sp(K)_{-N} ,$$

$$SO(N)_K \longleftrightarrow SO(K)_{-N} , \qquad O(N)^1_{K,K-1+L} \longleftrightarrow O(K)^1_{-N,-N+1+L} , \quad O(N)^0_{K,K} \longleftrightarrow Spin(K)_{-N} .$$
(1.1)

In the case of unitary gauge groups, these dualities were discussed in [1–6]. The dualities for $SO$ and $Sp$ gauge groups were derived in [7]. The $O(N)$ and $Spin(N)$ results were derived in [8].[1] To these, we will contribute the following identities:[2]

$$(E_8)_1 \longleftrightarrow \text{trivial} \longleftrightarrow SU(9)_1/\mathbb{Z}_3 , \qquad SU(6)_1 \longleftrightarrow (E_7)_1 \times SU(3)_{-1} , \quad SU(5)_1 \longleftrightarrow SU(5)_{-1} ,$$

$$(E_7)_1 \longleftrightarrow SU(2)_{-1} \longleftrightarrow SU(8)_1/\mathbb{Z}_2 , \quad SU(2)_3 \longleftrightarrow (E_7)_1 \times (F_4)_{-1} , \qquad SO(3)_8 \longleftrightarrow PSU(3)_{-6} ,$$

$$(E_6)_1 \longleftrightarrow SU(3)_{-1} \longleftrightarrow PSp(4)_1 , \qquad SU(2)_7 \longleftrightarrow (E_7)_1 \times (G_2)_{-2} , \qquad (G_2)_1 \longleftrightarrow (F_4)_{-1} .$$
(1.3)

An important feature of the dualities (1.3) (unlike some of the entries in (1.1) [6]) is that they are equivalences among bosonic TQFTs, i.e. they do not require a choice of spin structure. We can also combine (1.3) with some of the results of (1.1) to show that

$$(G_2)_1 \longleftrightarrow U(2)_{3,1} , \qquad (G_2)_2 \longleftrightarrow U(2)_{-7,-1} .$$
(1.4)

One way to summarize some of the dualities in (1.3) is via the equivalence

$$(E_N)_1 \longleftrightarrow SU(9-N)_{-1} .$$
(1.5)

In particular, as discussed in section 2 this naturally extends to all $0 \leq N \leq 8$ with a suitable identification of the group $E_N$ for small $N$.

The typical path to proving such dualities of Chern-Simons theories is to establish the equivalence of their corresponding chiral algebras. For connected and simply connected gauge groups, these are standard Kac-Moody current algebras (for a review see *e.g.* [9]). Gauging additional discrete symmetries, or reducing the gauge group by a central quotient modifies the chiral algebra respectively by an orbifold or an extension [10, 11].

One way to prove that two chiral algebras are equivalent is to start from a chiral algebra of a larger group at unit level (realized for a classical group for instance by suitable free 2$d$ fermions.) We then decompose the initial algebra under a conformally embedded product chiral algebra, with each factor related to a given Chern-Simons theory (see *e.g.* [9]). In section 2, we briefly review this procedure in the case of classical groups, and then we follow it for conformal embeddings of exceptional Kac-Moody algebras $(E_N)_1, (F_4)_1$, and $(G_2)_1$ studied in [12–15].

The outcome of our analysis are the dualities (1.3) and (1.4). In each case, we track the relative gravitational Chern-Simons counterterm across the duality. For some of these dualities,

---

[1]For some groups in (1.1), a careful definition of the Chern-Simons theory requires multiple levels. In the unitary group case the $SU(N)$ and $U(1)$ subalgebras can have separate integral levels [6] and we set

$$U(N)_{K,L} \equiv (SU(N)_K \times U(1)_{NL})/\mathbb{Z}_N .$$
(1.2)

Meanwhile for the orthogonal group there are two additional discrete levels indicating the action for the gauged reflection. In the notation above these are a superscript valued mod two and a second subscript valued mod eight [8].

[2]Our level conventions are $SO(3)_K \equiv SU(2)_{2K}/\mathbb{Z}_2$ , and $PSU(3)_K \equiv SU(3)_K/\mathbb{Z}_3$ .

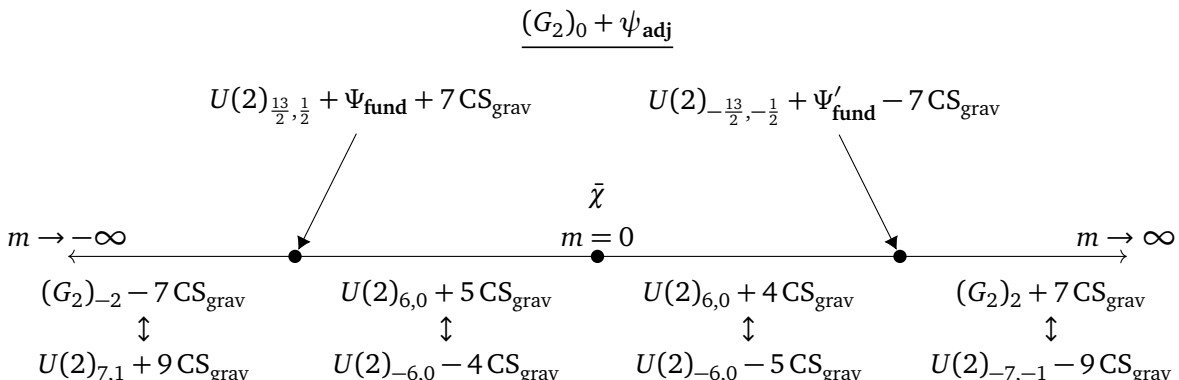

Figure 1: A conjectural phase diagram of $(G_2)_0$ coupled to an adjoint Majorana fermion as a function of the fermion mass $m$. When $m = 0$ the theory is time-reversal invariant, and has $\mathcal{N} = 1$ supersymmetry. At large $|m|$ the IR is described by a TQFT visible semiclassically. As $|m|$ is reduced there is a transition to a quantum phase. This transition is weakly coupled in dual variables with a $U(2)$ gauge field and a fundamental fermion. At $m = 0$ there is also massless fermion $\bar{\chi}$ which is the Goldstino arising from the expected spontaneous breaking of supersymmetry [30]. In each phase, we also indicate the coefficient of the gravitational Chern-Simons term which can be used as a consistency check on the phase diagram (see section 4).

we also describe in some detail the spins of the anyons as well as the map of lines between the two presentations of the theory.

In section 3 we apply our topological field theory analysis to conjecture several new boson-boson Chern-Simons-matter dualities.[3] They take the form:

$$(E_N)_1 + N_f \; \Phi_{\textbf{fund}} \qquad \longleftrightarrow \qquad SU(9 + N_f - N)_{-1} + N_f \; \Phi'_{\textbf{fund}} \,, \tag{1.6}$$

where in the above $1 \leq N \leq 8$, $N_f \leq N$, and the meaning of the fundamental representation of $E_N$ will be clarified below. As well as:

$$
\begin{aligned}
(E_6)_1 + \Phi_{\textbf{27}} \qquad &\longleftrightarrow \qquad (G_2)_{-1} + \Phi'_{\textbf{7}} \,, \\
Sp(3)_1 + \Phi_{\textbf{14}'} \qquad &\longleftrightarrow \qquad (G_2)_{-1} \times SU(3)_{-1} + \Phi'_{\textbf{3}} \,, \\
(F_4)_1 + \Phi_{\textbf{26}} \qquad &\longleftrightarrow \qquad (F_4)_{-1} + \Phi'_{\textbf{26}} \,,
\end{aligned}
\tag{1.7}
$$

where in the last line above, the content of the duality is that the IR theory has emergent time-reversal symmetry.

Our analysis parallels that of [6–8, 16, 17], which analyzed boson-fermion dualities for classical groups, extending previous large $N$ analysis in [18–20], connections to related supersymmetric dualities in [21–23], and recent work on particle vortex duality [24–27] building on the foundational results in [28, 29]. In each case of (1.6)-(1.7), relevant deformations of the proposed dualities yield equivalences among TQFTs which can be verified from our earlier results (1.3).

As a final application of our TQFT results, in section 4 we study the phase diagram of $(G_2)_0$ coupled to either an adjoint Majorana fermion, or two Majorana fermions in the fundamental **7**. Our results for the adjoint Majorana case are illustrated in Figure 1.

Phase diagrams of this sort have been investigated for classical groups in [8, 31–34]. As a function of the fermion masses there are semi-classical phases which are visible when $|m|$ is large. As $|m|$ is reduced there is a phase transition to a new *quantum phase* which is not obvious

---

[3]Using known boson-fermion dualities, we can also obtain fermionic duals of some of these theories.

from the UV Lagrangian. In our models, the transitions to the quantum phases are weakly coupled in dual variables with $U(2)$ gauge groups which are motivated by the TQFT dualities (1.4). The transition to the quantum phase is mediated by a $U(2)$ fundamental fermion dialing through zero mass.

In the case of $(G_2)_0$ coupled to an adjoint fermion, the quantum phase we discuss includes the special point $m = 0$ where the theory has $3d$ $\mathcal{N} = 1$ supersymmetry. Our results thus join a growing body of recent literature on such models [35–39]. For $(G_2)_0$ coupled to two fundamental fermions, the theory enjoys a $U(1)$ global symmetry and in the quantum phase this symmetry is spontaneously broken leading to a periodic scalar at long distances as in [32, 34, 40–43].

## 2 Duality of Chern-Simons Theories

### 2.1 Conformal Embeddings and Chiral Algebra Dualities

A conformal embedding of a $2d$ chiral algebra is the decomposition of a chiral algebra into smaller chiral algebras while respecting the conformal symmetry (see *e.g.* [9]). Such decompositions give rise to equivalences among chiral algebras and hence dualities of the corresponding Chern-Simons theories [11].

In general, one starts with a affine Kac-Moody algebra at unit level and decomposes it into a finite product of other Kac-Moody algebras.[4] Relatedly one obtains a decomposition of representations of the initial chiral algebra into modules of the product. To obtain a duality, it is essential that each chiral algebra factor acts faithfully and that all modules appear in the decomposition of some representation. To account for selection rules, one must often extend the chiral algebra. This is equivalent to performing a quotient on the gauge group in the Chern-Simons theory [10, 11].

Let us illustrate this idea briefly in several examples. The first is the decomposition of the chiral algebra $SU(NK)_1$ into $SU(N)_K \times SU(K)_N$ (see *e.g.* [44, 45]). All representations of $SU(N)_K$ and $SU(K)_N$ appear in the decomposition. Each representation of $SU(N)_K$ is paired with a representation of the coset $SU(NK)_1/SU(K)_N$. This gives a duality between chiral algebras. Using the map between cosets and Chern-Simons theories from [11] we find the Chern-Simons duality [45]

$$SU(N)_K \quad \longleftrightarrow \quad \frac{SU(NK)_1 \times SU(K)_{-N}}{\mathbb{Z}_K} \, . \tag{2.1}$$

This is an equivalence between bosonic TQFTs and does not require a spin structure. The quotient denotes a gauging of the diagonal $\mathbb{Z}_K$ one-form symmetry [46] (for a recent review see *e.g.* [47]), and it corresponds in $2d$ to extending the chiral algebras [10, 11]. This basic $2d$ chiral algebra duality can be generalized to a rich set of level-rank dualities discussed in [6].

Another example is the decomposition of the chiral algebra $Spin(NK)_1$, which represents $NK$ real fermions in $2d$ [8, 44, 48].

$$Spin(NK)_1 \supset Spin(N)_K \times Spin(K)_N \, . \tag{2.2}$$

Depending on $N$ and $K$, not every highest weight representation of the subalgebra $Spin(N)_K$ can appear in the decomposition of representations of $Spin(NK)_1$. If the representations of the subalgebra that appear only consist of tensorial representations (*i.e.* those that can be built from the vector representation), then we extend the chiral algebra to $SO(N)_K$ by including

---

[4]If the initial algebra is not at unit level, no such finite product decompositions exist [12, 13].

the simple current corresponding to the $\mathbb{Z}_2$ center in $Spin(N)$ that gives rise to the quotient $SO(N) = Spin(N)/\mathbb{Z}_2$. The conformal embedding then implies a duality between $Spin(N)_K$ (or $SO(N)_K$ if it is extended) and the coset $Spin(NK)_1/Spin(K)_N$ or $Spin(NK)_1/SO(K)_N$ depending on the representations of $Spin(K)$ that appear in the decomposition. This gives rise to dualities of the corresponding Chern-Simons theories [7,8] which are stated in (1.1).

In the rest of the section we will apply the above method to the conformal embeddings into the exceptional chiral algebras $(E_N)_1, (F_4)_1$ and $(G_2)_1$ which were studied in [12–15]. They are:

$$
\begin{array}{lll}
(E_8)_1 \supset SU(9)_1 \,, & (E_8)_1 \supset SU(2)_{16} \times SU(3)_6 \,, & (E_8)_1 \supset (E_7)_1 \times SU(2)_1 \,, \\
(E_8)_1 \supset SU(5)_1 \times SU(5)_1 \,, & (E_8)_1 \supset (E_6)_1 \times SU(3)_1 \,, & (E_8)_1 \supset (G_2)_1 \times (F_4)_1 \,, \\
(E_7)_1 \supset SU(8)_1 \,, & (E_7)_1 \supset Spin(12)_1 \times SU(2)_1 \,, & (E_7)_1 \supset SU(6)_1 \times SU(3)_1 \,, \\
(E_7)_1 \supset SU(2)_3 \times (F_4)_1 \,, & (E_7)_1 \supset (G_2)_1 \times Sp(3)_1 \,, & (E_7)_1 \supset SU(2)_7 \times (G_2)_2 \,, \\
(E_6)_1 \supset Sp(4)_1 \,, & (E_6)_1 \supset SU(6)_1 \times SU(2)_1 \,, & (E_6)_1 \supset SU(3)_2 \times (G_2)_1 \,, \\
(F_4)_1 \supset SU(3)_2 \times SU(3)_1 \,, & (F_4)_1 \supset Sp(3)_1 \times SU(2)_1 \,, & (F_4)_1 \supset (G_2)_1 \times SU(2)_8 \,, \\
(G_2)_1 \supset SU(2)_3 \times SU(2)_1 \,. & &
\end{array}
$$
$$\tag{2.3}$$

The decompositions for representations under these conformal embedding are given in [13].[5]

## 2.2 Exceptional Chern-Simons Dualities

From the conformal embeddings (2.3) and the corresponding decomposition of the representations [13] we find the following dualities between non-spin Chern-Simons theories

$$
\begin{array}{ccccc}
(E_8)_1 & \longleftrightarrow & \text{trivial} - 16\text{CS}_{\text{grav}} & \longleftrightarrow & SU(9)_1/\mathbb{Z}_3 \,, \\
(E_7)_1 & \longleftrightarrow & SU(2)_{-1} - 16\text{CS}_{\text{grav}} & \longleftrightarrow & SU(8)_1/\mathbb{Z}_2 \,, \\
(E_6)_1 & \longleftrightarrow & SU(3)_{-1} - 16\text{CS}_{\text{grav}} & \longleftrightarrow & PSp(4)_1 \,, \\
SU(5)_1 & \longleftrightarrow & SU(5)_{-1} - 16\text{CS}_{\text{grav}} \,, & & \\
SU(6)_1 & \longleftrightarrow & (E_7)_1 \times SU(3)_{-1} \,, & & \\
SO(3)_8 & \longleftrightarrow & PSU(3)_{-6} - 16\text{CS}_{\text{grav}} \,, & & \\
(G_2)_1 & \longleftrightarrow & (F_4)_{-1} - 16\text{CS}_{\text{grav}} \,, & & \\
SU(2)_3 & \longleftrightarrow & (E_7)_1 \times (F_4)_{-1} \,, & & \\
SU(2)_7 & \longleftrightarrow & (E_7)_1 \times (G_2)_{-2} \,. & &
\end{array}
$$
$$\tag{2.5}$$

In the above we have included the relative value of the gravitational Chern-Simons term $\text{CS}_{\text{grav}}$ across each duality. Our normalization is such that $\text{CS}_{\text{grav}} = \pi \int_{4d} \hat{A}(R)$. When the coefficient is a multiple of 16 (as in all cases above), this counterterm does not require a choice of spin structure. Hence $(-16)\text{CS}_{\text{grav}}$ defines an invertible non-spin TQFT with framing anomaly $c = 8$ that has only the trivial line, such as the $(E_8)_1$ Chern-Simons theory.[6]

---

[5]In the above list we include only the conformal embeddings where we can identify all the chiral algebras yielding the representations that appear in the decompositions. Examples of conformal embeddings not listed above are:
$$
(E_6)_1 \supset SU(3)_9 \,, \quad (E_6)_1 \supset (G_2)_3 \,, \quad (E_7)_1 \supset SU(3)_{21} \,, \quad (E_8)_1 \supset Sp(2)_{12} \,. \tag{2.4}
$$

[6]In general, the invertible spin TQFT $SO(L)_1$ has partition function $\exp\left(-iL\text{CS}_{\text{grav}}\right)$ (see *e.g.* Appendix B in [27]). The theory has framing anomaly $c = L/2$. Since $8 \int_{4d} \hat{A}(R) \in \mathbb{Z}$ on any closed orientable four-manifold, for $L \in 16\mathbb{Z}$ the gravitational Chern-Simons term does not require a spin structure.

Let us comment on some aspects of these dualities:

- The theories $(E_N)_1$ and $SU(N)_1$ are Abelian TQFTs.[7] Thus, the dualities in (2.5) that involve only $(E_N)_1$ and $SU(N)_1$ can be alternatively obtained from the defining properties of Abelian TQFTs. Namely, an Abelian TQFT is specified by the fusion rules, the spin of the lines, and the framing anomaly (for a review see *e.g.* [49]).

- The duality $SU(5)_1 \longleftrightarrow SU(5)_{-1} - 16\text{CS}_{\text{grav}}$ implies the theory $SU(5)_1$ is time-reversal invariant as a non-spin TQFT. Using $SU(5)_1 \longleftrightarrow SO(5)_2 \cong Sp(2)_2/\mathbb{Z}_2$, this also follows from the time-reversal symmetry of $Sp(2)_2$ [7]. This duality was also discussed in [50].

- The last three dualities in (2.5) give examples of a phenomenon recently discussed in [47]. Specifically, they present factorizations of TQFTs into the product of a "minimal Abelian TQFT" [47] and another TQFT. The factorization is a consequence of the one-form global symmetry [47].

One way to summarize the first five dualities in (2.5) is[8]

$$(E_N)_1 \quad \longleftrightarrow \quad SU(9-N)_{-1} - 16\text{CS}_{\text{grav}} \,, \tag{2.6}$$

where $E_5 \cong Spin(10), E_4 \cong SU(5)$, and $E_3 \cong SU(3) \times SU(2)$. This extends to integers $0 \le N \le 8$ provided that we define[9]

$$(E_2)_1 \equiv U(2)_{1,7} \,, \quad (E_1)_1 \equiv \frac{SU(2)_1 \times (\mathbb{Z}_4)_{14}}{\mathbb{Z}_2} \,, \quad (\widetilde{E}_1)_1 \equiv U(1)_8 \,, \qquad (E_0)_1 \equiv (\mathbb{Z}_3)_2 \,. \tag{2.7}$$

In the above, we use the convention that $(\mathbb{Z}_n)_k$ is the $\mathbb{Z}_n$ topological gauge theory that can be expressed using $U(1) \times U(1)$ gauge fields $x, y$ with the action $\frac{k}{4\pi} x dx + \frac{n}{2\pi} x dy$. Moreover, the $\mathbb{Z}_2$ quotient in $(E_1)_1$ on $(\mathbb{Z}_4)_{14}$ uses the line $\exp(3i \oint x + 2i \oint y)$. Finally, the two theories $(E_1)_1$ and $(\widetilde{E}_1)_1$ are both dual to $SU(8)_{-1}$ (and hence dual to each other). We list them both here for later application to dualities involving matter in section 3.

Apart from our unusually careful attention to the global form of the group, the above definition of $E_N$ for small $N$ are standard and natural from e.g. the perspective of Higgsing.[10] This will play a crucial role in our discussion of Chern-Simons-matter dualities in section 3. We can also use the level-rank duality [6] to dualize the right-hand side of (2.6) to obtain an equality of spin TQFTs

$$\text{spin TQFTs:} \quad (E_N)_1 \quad \longleftrightarrow \quad U(1)_{-N+9} - 2(N-1)\text{CS}_{\text{grav}} \,, \tag{2.8}$$

where on both sides we tensor with the theory $\{1, \psi\}$ ($\psi$ is the transparent spin 1/2 line) with zero framing anomaly to turn them into spin TQFTs.[11]

Finally, let us remark that the last two dualities in (2.5) imply that

$$(F_4)_{-1} \longleftrightarrow \left(SU(2)_3 \times (E_7)_{-1}\right)/\mathbb{Z}_2 \,, \qquad (G_2)_{-2} \longleftrightarrow \left(SU(2)_7 \times (E_7)_{-1}\right)/\mathbb{Z}_2 \,. \tag{2.9}$$

---

[7]They can be described as Abelian Chern-Simons theories with K-matrices given by their Cartan matrices.

[8]The duality (2.6) with $N = 5$ is a special case of the duality $Spin(L)_1 \longleftrightarrow Spin(16-L)_{-1} - 16\text{CS}_{\text{grav}}$ using $SU(4) \cong Spin(6)$, and we include it in (2.6) for completeness. This duality for general $L$ can be derived from the series of conformal embeddings $Spin(L)_1 \times Spin(16-L)_1 \subset Spin(16)_1 \subset (E_8)_1$ [12]. This "16 periodicity" for the TQFT described by $Spin(L)_1$ is also discussed in [51].

[9]Since the theories $(E_N)_1$ and $SU(9-N)_{-1}$ are Abelian TQFTs, the dualities can be proven from the fusion rules and the spin of the lines.

[10]For instance they also occur in the discussion of five-dimensional superconformal field theories [52, 53]. The disconnected parts of $E_1$ and $E_0$ also exist in the global symmetries of the corresponding superconformal field theories, which are the subgroups preserving the $(p, q)$ brane webs engineering these theories.

[11]For $N = 1$ the above duality is also valid between non-spin TQFTs [7, 54].

Using $(F_4)_{-1} \leftrightarrow (G_2)_1 + 16\text{CS}_{\text{grav}}$, $(E_7)_1 \leftrightarrow SU(2)_{-1} - 16\text{CS}_{\text{grav}}$ in (2.5) and $SU(2)_1 \leftrightarrow U(1)_2$ we can further simplify them into the non-spin dualities

$$(G_2)_1 \quad\longleftrightarrow\quad U(2)_{3,1}, \qquad (G_2)_2 \quad\longleftrightarrow\quad U(2)_{-7,-1} - 16\text{CS}_{\text{grav}}. \qquad (2.10)$$

We will make use of these TQFT dualities in section 4.

## 2.3 Examples

The map of representations/lines across the dualities (2.5) can be obtained from the decomposition of modules in the conformal embeddings [13]. Here we discuss some of them.

Consider the duality

$$(E_N)_1 \quad\longleftrightarrow\quad SU(9-N)_{-1} - 16\text{CS}_{\text{grav}}, \qquad (2.11)$$

where for $5 \geq N \geq 0$ the left-hand side is given in (2.7). The lines on the left-hand side are labelled by the Dynkin labels $\lambda_1, \lambda_2, \cdots, \lambda_n$, while the lines on the right-hand side are labelled by the antisymmetric tensor representation with $r$ indices with $r = 0, 1, \cdots, (8-N)$ mod $(9-N)$.

- $N = 8$. The highest weight representations in $(E_8)_1$ satisfy

$$2\lambda_1 + 3\lambda_2 + 4\lambda_3 + 5\lambda_4 + 6\lambda_5 + 4\lambda_6 + 2\lambda_7 + 3\lambda_8 \leq 1, \qquad (2.12)$$

and thus the Chern-Simons theory only has the trivial line, which maps to the trivial line on the other side of the duality.

- $N = 7$. The highest weight representations in $(E_7)_1$ satisfy

$$2\lambda_1 + 3\lambda_2 + 4\lambda_3 + 3\lambda_4 + 2\lambda_5 + \lambda_6 + 2\lambda_7 \leq 1, \qquad (2.13)$$

and thus there are two lines, with $\lambda_6 = 0, 1$ and all other Dynkin labels vanishing. The two lines map to $r = 0, 1$ on the right-hand side of the duality. The lines have spin $h[r] = \frac{3}{4}r^2$ mod 1.

- $N = 6$. The highest weight representations in $(E_6)_1$ satisfy

$$\lambda_1 + 2\lambda_2 + 3\lambda_3 + 2\lambda_4 + \lambda_5 + 2\lambda_6 \leq 1, \qquad (2.14)$$

and thus there are three lines, with $(\lambda_1, \lambda_5) = (0,0), (0,1), (1,0)$ and all other Dynkin labels vanishing. The three lines map to $r = 0, 1, 2$ on the right-hand side of the duality. The lines have spin $h[r] = \frac{2}{3}r^2$ mod 1.

- $N = 5$. The highest weight representations in $(E_5)_1 \cong Spin(10)_1$ satisfy

$$\lambda_1 + 2\lambda_2 + 2\lambda_3 + \lambda_4 + \lambda_5 \leq 1, \qquad (2.15)$$

and thus there are 4 lines, with $(\lambda_1, \lambda_4, \lambda_5) = (0,0,0), (0,1,0), (1,0,0), (0,0,1)$ and all other Dynkin labels vanishing. The three lines map to $r = 0, 1, 2, 3$ on the right-hand side of the duality. The lines have spin $h[r] = \frac{5}{8}r^2$ mod 1.

- $N = 4$. The highest weight representations in $(E_4)_1 \cong SU(5)_1$ are antisymmetric tensor representations with $r = 0, 1, 2, 3, 4$ indices. There are 5 lines, and they map to the representations in $SU(5)_{-1}$ by

$$r \longrightarrow r' = 2r. \qquad (2.16)$$

This map has inverse $r = 3r'$. The lines have spins $h[r] = \frac{2}{5}r^2$ mod 1.

- $N = 3$. The lines in $(E_3)_1 \cong SU(3)_1 \times SU(2)_1$ theory can be labelled by antisymmetric tensor representations in $SU(3)$ and $SU(2)$ with $r_1$ and $r_2$ indices, with $r_1 = 0, 1, 2$ and $r_2 = 0, 1$. There are 6 lines, and they map to the lines in $SU(6)_1$ theory as $r$-index antisymmetric tensor representations with $r = 2r_1 + 3r_2 \mod 6$. The lines have spins $h[r] = \frac{7}{12} r^2 \mod 1$.

- $N = 2$. The lines in $(E_2)_1 \equiv U(2)_{1,7}$ can be labelled by $SU(2)$ isospin zero and $U(1)$ charge $Q = 2q = 0, 2, 4, 6, 8, 10, 12$, and thus there are 7 lines. They map to the anti-symmetric tensor representations of $SU(7)$ with $r = 3q \pmod 7$ indices. The lines have spins $h[r] = \frac{4}{7} r^2 \mod 1$.

- $N = 1$. The lines in $(E_1)_1 \equiv \big( SU(2)_1 \times (\mathbb{Z}_4)_{14} \big) / \mathbb{Z}_2$ can be labelled by powers $p$ of the basic magnetic line $\exp(i \oint y)$ in $(\mathbb{Z}_4)_{14}$, which has order 8. They map to the antisymmetric tensor representations of $SU(8)$ with $p \pmod 8$ indices. The lines have spins $h[p] = \frac{9}{16} p^2 \mod 1$.

- $N = 0$. The lines in $(E_0)_1 \equiv (\mathbb{Z}_3)_2$ can be labelled by powers $p$ of the basic magnetic line $\exp(i \oint y)$, which has order 9. They map to the antisymmetric tensor representations of $SU(9)$ with $r = 4p \pmod 9$ indices. The spins are $h[r] = \frac{5}{9} r^2 \mod 1$.

Consider the duality

$$SO(3)_8 \quad \longleftrightarrow \quad PSU(3)_{-6} - 16\mathrm{CS}_{\mathrm{grav}} \,. \tag{2.17}$$

The representations on the right have $SU(2)$ isospin $j = 0, 1, 2, 3, 4_\pm$ where the subscript is associated with the $\mathbb{Z}_2$ quotient of $SO(3)_8 = SU(2)_{16}/\mathbb{Z}_2$. There are 6 lines, and they map to the $PSU(3)_{-6}$ theory as the lines with $SU(3)$ Dynkin labels

$$(\lambda_1, \lambda_2) = (0,0), (2,2)_0, (3,0), (1,1), (2,2)_1, (2,2)_2 \,, \tag{2.18}$$

where the subscript denotes the 3 copies associated with the $\mathbb{Z}_3$ quotient in $PSU(3)_6 = SU(3)_6/\mathbb{Z}_3$. The lines have spins $h = 0, \frac{1}{9}, \frac{1}{3}, \frac{2}{3}, \frac{1}{9}, \frac{1}{9} \mod 1$. The theory has $S_3$ 0-form symmetry that permutes the 3 copies associated with the $\mathbb{Z}_3$ quotient.[12]

Consider the duality

$$(G_2)_1 \quad \longleftrightarrow \quad (F_4)_{-1} - 16\mathrm{CS}_{\mathrm{grav}} \,. \tag{2.20}$$

The highest weigh representations for $(G_2)_1$ satisfy $2\lambda_1 + \lambda_2 \leq 1$ and thus there are two lines, with $\lambda_2 = 0, 1$ and $\lambda_1 = 0$. The highest weight representations for $(F_4)_1$ satisfy $2\lambda_1' + 3\lambda_2' + 2\lambda_3' + \lambda_4' \leq 1$, and thus the theory also has two lines, with $\lambda_4' = 0, 1$ and all other Dynkin labels vanish. The map of the duality is then $\lambda_4' = \lambda_2$. The lines have spins $h = 0, \frac{2}{5} \mod 1$.

Consider the duality

$$(G_2)_1 \quad \longleftrightarrow \quad U(2)_{3,1} \,. \tag{2.21}$$

---

[12]The left-hand side of the duality (2.17) has another description [7]

$$SO(3)_8 \quad \longleftrightarrow \quad \frac{Spin(24)_1 \times Spin(8)_{-3}}{\mathbb{Z}_2 \times \mathbb{Z}_2} \quad \longleftrightarrow \quad \frac{Spin(8)_1 \times Spin(8)_{-3}}{\mathbb{Z}_2 \times \mathbb{Z}_2} - 16\mathrm{CS}_{\mathrm{grav}} \,, \tag{2.19}$$

where the $\mathbb{Z}_2 \times \mathbb{Z}_2$ quotient uses the diagonal center, and the second duality uses $Spin(24)_1 \longleftrightarrow Spin(8)_1 - 16\mathrm{CS}_{\mathrm{grav}}$. The TQFT consists of the lines in $Spin(8)_{-3}$ that are invariant under the $\mathbb{Z}_2 \times \mathbb{Z}_2$ one-form symmetry. The lines in $Spin(8)_{-3}$ that generate the one-form symmetry have Dynkin labels $(0,0,0,0)$, $(3,0,0,0)$, $(0,0,0,3)$ and $(0,0,3,0)$, which form closed orbits under the $Spin(8)$ triality. Thus the triality is a symmetry of the TQFT, and it reproduces the 0-form permutation symmetry.

The representations on the right-hand side can be labelled by zero $U(1)$ charge and $SU(2)$ isospin $j = 0, 1$. These two lines map to the lines $\lambda_2 = 0, 1$ in $(G_2)_1$. We remark that the dualities (2.20),(2.21) provide different Chern-Simons theory descriptions for the Fibonacci Anyons (see *e.g.* [55–57]), namely, the only non-trivial line $\tau$ in the theory has the fusion rule $\tau \times \tau = 1 + \tau$.

Consider the duality

$$(G_2)_2 \quad \longleftrightarrow \quad U(2)_{-7,-1} - 16\text{CS}_{\text{grav}} \,. \tag{2.22}$$

The highest weight representations in $(G_2)_2$ satisfy $2\lambda_1 + \lambda_2 \leq 2$ and thus there are four lines, with $(\lambda_1, \lambda_2) = (0, 0), (0, 2), (0, 1), (1, 0)$. They map to the lines on the right-hand side of $U(1)$ charge zero and $SU(2)$ isospin $j = 0, 1, 2, 3$. The lines have spins $h = 0, \frac{7}{9}, \frac{1}{3}, \frac{2}{3}$ mod 1.

# 3  Chern-Simons-Matter Dualities

In this section we conjecture new Chern-Simons-matter dualities. They enjoy the consistency check that under relevant deformations, they flow to the rigorous dualities of pure Chern-Simons theories discussed in section 2. In particular, this also implies that all anomalies of global symmetries match across our dualities. We can also verify that the gravitational Chern-Simons counterterms in the dualities are reproduced in a non-trivial way from integrating out massive fermions

The Chern-Simons-matter theories that participate in the dualities have small Chern-Simons levels and small gauge groups. Thus the dualities discussed here do not admit an obvious semi-classical limit, unlike the dualities proposed for other gauge groups such as in [6, 7]. (It is perhaps possible that they can be obtained from supersymmetric dualities as in [21–23].)

Throughout, we assume that theories involving scalars are driven to fixed points by a suitable potential. Moreover we often have need of various potentials achieving desired patterns of Higgsing. The general existence of these potentials is discussed in Appendix A.

We adopt the convention that $\Phi$ is a complex scalar, and $\Psi$ is a complex (Dirac) fermion. We also set $m_{L,R}$ to be the relevant deformation parameter on the left and right-hand side of the dualities.

The dualities proposed below involve complex scalars or Dirac fermions, and the theories have (at least) a $U(1)$ global symmetry. In particular we assume that all scalar potentials respect the global phase rotation $\Phi \to e^{i\alpha}\Phi$. We check that the dualities are consistent under relevant deformations preserving this symmetry.

## 3.1  $Sp(3) \leftrightarrow SU(3)$ **or** $U(1)$

We propose that the following theories are dual (IR equivalent):

$$
\begin{aligned}
Sp(3)_1 + \Phi_{\mathbf{14'}} \quad &\longleftrightarrow \quad (G_2)_{-1} \times SU(3)_{-1} + \Phi'_{\mathbf{3}} - 16\text{CS}_{\text{grav}} \\
&\longleftrightarrow \quad (G_2)_{-1} \times U(1)_{\frac{5}{2}} + \Psi - 11\text{CS}_{\text{grav}} \,,
\end{aligned}
\tag{3.1}
$$

where the second line uses the $SU/U$ duality discussed in [6] with $\Psi$ a Dirac fermion of charge one.

Note that $Sp(3)$ has a $U(3)$ subgroup and the index of the $SU(3)$ subgroup is two. Under $U(3)$ the $\mathbf{14'}$ decomposes as

$$\mathbf{14'} \to \mathbf{1}_3 \oplus \mathbf{1}_{-3} \oplus \mathbf{6}_{-1} \oplus \bar{\mathbf{6}}_1 \,. \tag{3.2}$$

We assume that on the left-hand side of (3.1) the potential is such that for one sign of the relevant operator an $SU(3)$ singlet gets an expectation value. (The unbroken $\mathbb{Z}_3$ in $U(1)$ is in fact part of the $SU(3)$).

With this setup we can now investigate the massive phases. The fixed point has a relevant operator described by the mass terms in the dual descriptions. The potentials in the scalar theories are such that the mass deformations lead to the following pure Chern-Simons dualities in (2.5):

$$m_L^2 > 0 \longleftrightarrow m_R^2 < 0 : \qquad Sp(3)_1 \longleftrightarrow (G_2)_{-1} \times SU(2)_{-1} - 16\mathrm{CS}_{\mathrm{grav}}\,, \qquad (3.3)$$
$$m_L^2 < 0 \longleftrightarrow m_R^2 > 0 : \qquad SU(3)_2 \longleftrightarrow (G_2)_{-1} \times SU(3)_{-1} - 16\mathrm{CS}_{\mathrm{grav}}\,.$$

Let us explain in more detail how to obtain the dualities (3.3) from (2.5). To compare the top line of (3.3) with (2.5), we use the two spin dualities $Sp(3)_1 \longleftrightarrow Sp(1)_{-3} - 12\mathrm{CS}_{\mathrm{grav}} \cong SU(2)_{-3} - 12\mathrm{CS}_{\mathrm{grav}}$, $SU(2)_{-1} \cong Sp(1)_{-1} \longleftrightarrow Sp(1)_1 + 4\mathrm{CS}_{\mathrm{grav}} \cong SU(2)_1 + 4\mathrm{CS}_{\mathrm{grav}}$ [7].[13] The second duality in (3.3) is equivalent to $(G_2)_{-1}$ being the non-Abelian sector of $SU(3)_2$, namely, $(G_2)_{-1} \longleftrightarrow (SU(3)_2 \times SU(3)_1)/\mathbb{Z}_3 + 16\mathrm{CS}_{\mathrm{grav}}$ [47]. The later follows from (2.10) and the two spin dualities $U(2)_{-3,-1} \longleftrightarrow U(3)_{2,-1} + 10\mathrm{CS}_{\mathrm{grav}}$, $U(1)_{-3} \longleftrightarrow SU(3)_1 + 6\mathrm{CS}_{\mathrm{grav}}$ [6].

## 3.2 Time-Reversal Invariant $F_4$ Theory

We conjecture the IR duality

$$(F_4)_1 + \Phi_{\mathbf{26}} \qquad \longleftrightarrow \qquad (F_4)_{-1} + \Phi'_{\mathbf{26}} - 16\mathrm{CS}_{\mathrm{grav}}\,. \qquad (3.4)$$

Observe that the two theories above are equivalent up to an action of time-reversal symmetry. Thus the claim in the duality is that the IR theory is time-reversal invariant.

Note that $F_4$ has a $G_2 \times SO(3)$ subgroup of index (1,4). Under this subgroup, the $\mathbf{26}$ decomposes as

$$\mathbf{26} \to (\mathbf{7},\mathbf{3}) \oplus (\mathbf{1},\mathbf{5})\,. \qquad (3.5)$$

We choose the potential on both sides such that for one sign of the relevant operator, a field in the $(\mathbf{1},\mathbf{5})$ gets a generic expectation value.[14] This Higgses $F_4$ to $G_2$ (there is no additional discrete subgroup by our assumption of a generic expectation value.)

In the gapped phases we find dualities from (2.5):

$$m_L^2 > 0 \longleftrightarrow m_R^2 < 0 : \qquad (F_4)_1 \longleftrightarrow (G_2)_{-1} - 16\mathrm{CS}_{\mathrm{grav}}\,, \qquad (3.6)$$
$$m_L^2 < 0 \longleftrightarrow m_R^2 > 0 : \qquad (G_2)_1 \longleftrightarrow (F_4)_{-1} - 16\mathrm{CS}_{\mathrm{grav}}\,.$$

## 3.3 $E_6 \longleftrightarrow G_2$

We conjecture the IR duality

$$(E_6)_1 + \Phi_{\mathbf{27}} \qquad \longleftrightarrow \qquad (G_2)_{-1} + \Phi'_{\mathbf{7}} - 16\mathrm{CS}_{\mathrm{grav}}\,. \qquad (3.7)$$

Note that $E_6$ has an $F_4$ subgroup of index 1. Under this subgroup, the $\mathbf{27}$ decomposes as

$$\mathbf{27} \to \mathbf{1} \oplus \mathbf{26}\,. \qquad (3.8)$$

---

[13]We also use the fact that a spin duality implies a non-spin duality when the sectors other than the factorized $\{1,\psi\}$ are non-spin TQFTs, and their framing anomaly differs by a multiple of 8 (*i.e.* the coefficients of the gravitational Chern-Simons term on the two sides differ by a multiple of 16) [54].

[14]By generic, we mean that $\Phi$ and $\bar{\Phi}$ do not commute as matrices in the $\mathbf{5}$ (the symmetric traceless tensor).

We choose the potential such that for one sign of the relevant operator, a field in the **1** gets an expectation value. We also use the fact that $G_2$ has an $SU(3)$ subgroup of index 1. Under $SU(3)$ the **7** decomposes as

$$7 \rightarrow \mathbf{1} \oplus \mathbf{3} \oplus \bar{\mathbf{3}} \,. \tag{3.9}$$

We chose the potential such that for one sign of the relevant operator, a field in the **1** gets an expectation value.

In the gapped phases we find the following dualities from (2.5):

$$
\begin{aligned}
m_L^2 > 0 \longleftrightarrow m_R^2 < 0 : \quad & (E_6)_1 \longleftrightarrow SU(3)_{-1} - 16\text{CS}_{\text{grav}} \,, \\
m_L^2 < 0 \longleftrightarrow m_R^2 > 0 : \quad & (F_4)_1 \longleftrightarrow (G_2)_{-1} - 16\text{CS}_{\text{grav}} \,.
\end{aligned}
\tag{3.10}
$$

## 3.4 $E_N \longleftrightarrow SU$ or $U(1)$

We conjecture the IR duality

$$(E_N)_1 + \Phi_{\mathbf{fund}} \quad \longleftrightarrow \quad SU(10-N)_{-1} + \Phi'_{\mathbf{fund}} - 16\text{CS}_{\text{grav}} \,, \tag{3.11}$$

where the allowed range of $N$ is $N \geq 3$ (it will be extended below), and the definition of the group $E_N$ for general $N$ is as specified in (2.7). Meanwhile, the definition of the fundamental representation of $E_N$ as follows:

$$E_8 : \mathbf{248} \,, \quad E_7 : \mathbf{56} \,, \quad E_6 : \mathbf{27} \,, \quad E_5 \cong Spin(10) : \mathbf{16} \,, \tag{3.12}$$

$$E_4 \cong SU(5) : \mathbf{10} \,, \quad E_3 \cong SU(3) \times SU(2) : (\mathbf{3}, \mathbf{2}) \,.$$

As we will see below, the essential property of this definition of the fundamental of $E_N$ is that a generic expectation value of such a fundamental can Higgs $E_N$ to $E_{N-1}$.

The duality sequence (3.11) can be extended to smaller $N$ as follows. For $N = 2$ we have two dualities:

$$(E_2)_1 \cong U(2)_{1,7} + \mathbf{2_3} \quad \longleftrightarrow \quad (E_2)_1 \cong U(2)_{1,7} + \mathbf{1_4} \quad \longleftrightarrow \quad SU(8)_{-1} + \Phi'_{\mathbf{fund}} - 16\text{CS}_{\text{grav}} \,. \tag{3.13}$$

Meanwhile for $N = 1$, there is an analogous duality involving $\widetilde{E}_1$ (but none for $E_1$):[15]

$$(\widetilde{E}_1)_1 \cong U(1)_8 + \mathbf{1_3} \quad \longleftrightarrow \quad SU(9)_{-1} + \Phi'_{\mathbf{fund}} - 16\text{CS}_{\text{grav}} \,. \tag{3.14}$$

In all of the dualities above, for one sign of the relevant operator, the scalar $\Phi$ on the left-hand side does not condense but the scalar $\Phi'$ condenses and Higgses $SU(10-N)$ to $SU(9-N)$. This reproduces the duality (2.6). For the other sign of the relevant operator, the scalar $\Phi$ condenses but the scalar $\Phi'$ does not. The left-hand side becomes a pure Chern-Simons theory, and we will check whether the theory reduces to $(E_{N-1})_1$ after Higgsing.

- For $N = 8$ the representation **248** decomposes under $E_8 \supset (E_7 \times SU(2))/\mathbb{Z}_2$ as

$$\mathbf{248} \rightarrow (\mathbf{1}, \mathbf{3}) \oplus (\mathbf{56}, \mathbf{2}) \oplus (\mathbf{133}, \mathbf{1}) \,. \tag{3.15}$$

  Thus a generic expectation value for $(\mathbf{1}, \mathbf{3})$ breaks the group to $E_7$. (Here we assume that real and imaginary parts of the vev are not aligned so that $SU(2)$ is broken.)

---

[15] This is similar to the pattern of relevant deformations of five-dimensional superconformal field theories with $E_N$ global symmetry [52, 53].

- For $N = 7$ the representation **56** decomposes under $E_7 \rightarrow (E_6 \times U(1))/\mathbb{Z}_3$ as

$$\mathbf{56} \rightarrow \mathbf{1}_3 \oplus \mathbf{1}_{-3} \oplus \mathbf{27}_{-1} \oplus \overline{\mathbf{27}}_1 \,. \tag{3.16}$$

  Thus condensing $\mathbf{1}_3$ breaks the group to $E_6$.

- For $N = 6$ the representation **27** decomposes under $E_6 \rightarrow (Spin(10) \times U(1))/\mathbb{Z}_4$ as

$$\mathbf{27} \rightarrow \mathbf{1}_{-4} \oplus \mathbf{10}_2 \oplus \mathbf{16}_{-1} \,. \tag{3.17}$$

  Thus condensing $\mathbf{1}_{-4}$ breaks the group to $Spin(10) = E_5$.[16]

- For $N = 5$ the representation **16** decomposes under $Spin(10) \rightarrow (SU(5) \times U(1))/\mathbb{Z}_5$ as

$$\mathbf{16} \rightarrow \mathbf{1}_{-5} \oplus \bar{\mathbf{5}}_3 \oplus \mathbf{10}_{-1} \,. \tag{3.18}$$

  Thus condensing $\mathbf{1}_{-5}$ breaks the group to $SU(5) = E_4$.

- For $N = 4$ the representation **10** decomposes under $SU(5) \rightarrow (SU(3) \times SU(2) \times U(1))/\mathbb{Z}_6$ as

$$\mathbf{10} \rightarrow (\mathbf{1}, \mathbf{1})_{-6} \oplus (\bar{\mathbf{3}}, \mathbf{1})_4 \oplus (\mathbf{3}, \mathbf{2})_{-1} \,. \tag{3.19}$$

  Thus condensing $(\mathbf{1}, \mathbf{1})_{-6}$ breaks the group to $SU(3) \times SU(2) = E_3$.

- For $N = 3$ condensing the scalar in $(\mathbf{3}, \mathbf{2})$ Higgses the gauge group to $U(2)$. To see the level, one can consider the Cartan subgroup embedded in $SU(2) \times SU(3)$ as $\mathrm{diag}(e^{i\alpha}, e^{-i\alpha}) \times \mathrm{diag}(e^{i\beta}, e^{i\gamma}, e^{-i\beta-i\gamma})$ with $\alpha, \beta, \gamma \in \mathbb{R}/(2\pi\mathbb{Z})$. Then the $SU(2)_1 \times SU(3)_1$ Chern-Simons term becomes $U(1)_2 \times (U(1)_2 \times U(1)_6)/\mathbb{Z}_2$. After condensing the scalar it becomes $\left(U(1)_2 \times U(1)_{14}\right)/\mathbb{Z}_2$, where $U(1)_2$ is embedded in $SU(2)$ from the original $SU(3)$ gauge group. Thus the Chern-Simons term after Higgsing is $U(2)_{1,7}$.

- For the first duality in the $N = 2$ case, condensing the scalar of charge 4 Higgses $(E_2)_1 \cong U(2)_{1,7} \cong \frac{SU(2)_1 \times U(1)_{14}}{\mathbb{Z}_2}$ to $\left(SU(2)_1 \times (Z_4)_{14}\right)/\mathbb{Z}_2 \cong (E_1)_1$ as expected. For the second duality, condensing $\mathbf{2}_3$ Higgses $U(2)$ down to a $U(1)$ subgroup embedded inside Cartan of $U(2)$. To see the level, we parametrize the Caratan of $U(2)$ as $\mathrm{diag}(e^{i\alpha+i\beta}, e^{i\alpha-i\beta})$. The condensation forces $3\alpha = \beta$. Since the Chern-Simons term for $U(2)_{1,7}$ is $\frac{\pi}{4}(\mathrm{Tr}(udu) + 3\,\mathrm{Tr}(u)d\,\mathrm{Tr}(u))$ with $U(2)$ gauge field $u$, by restricting it to the $U(1)$ subgroup specified by $3\alpha = \beta$ we get $U(1)_8 \cong (\widetilde{E}_1)_1$.

- For $N = 1$, condensing the scalar of charge 3 Higgses $(\widetilde{E}_1)_1 \cong U(1)_8$ down to $(\mathbb{Z}_3)_8 \cong (\mathbb{Z}_3)_2 \cong (E_0)_1$. (Here we use the fact that the level of $\mathbb{Z}_3$, defined below (2.7), is periodic mod 6 as a non-spin TQFT by the field redefinition $y \rightarrow y + x$.)

We can generalize the dualities above to allow $N_f$ scalars for $N_f \leq N$,

$$(E_N)_1 + N_f \, \Phi_{\mathbf{fund}} \quad \longleftrightarrow \quad SU(9 + N_f - N)_{-1} + N_f \, \Phi'_{\mathbf{fund}} - 16\,\mathrm{CS}_{\mathrm{grav}} \,. \tag{3.20}$$

For the case with $N = 1$, in the left hand side we use $\widetilde{E}_1$ and use the duality (3.14). We can also dualize the right-hand side to $U(1)_{-N+9+N_f/2}$ theory with $N_f$ fermions [6] and obtain the following boson-fermion duality

$$(E_N)_1 + N_f \, \Phi_{\mathbf{fund}} \quad \longleftrightarrow \quad U(1)_{-N+9+N_f/2} + N_f \, \Psi_{\mathbf{fund}} - 2\left(N - N_f/2 - 1\right)\mathrm{CS}_{\mathrm{grav}} \,. \tag{3.21}$$

The right-hand side has a $U(1)$ magnetic global symmetry, which is dual to the phase rotation of the complex scalar field in the $E_N$ gauge theory.

---

[16]Note that although the UV $E_6$ theory has the same matter content as that in Section 3.3 the potentials are different. Thus we do not expect that the theories discussed here and in Section 3.3 are dual.

# 4 Phase Diagrams of $(G_2)_0$ with Fermions

In this section we propose phase diagrams for theories involving $G_2$ gauge fields coupled with fermions. In the first subsection we study $(G_2)_0$ gauge theory with a Majorana fermion in the adjoint representation, while the second subsection we study $(G_2)_0$ gauge theory with two Majorana fermions in the fundamental **7** representation.

## 4.1 $(G_2)_0$ with a single adjoint fermion

Here we consider the theory $(G_2)_0$ with a Majorana adjoint $\psi_{\mathbf{adj}}$. We propose a phase diagram of this theory as a function of the mass $m$ of the adjoint. Our analysis is similar to that of [8, 32–34]. The phase structure is motivated by the TQFT duality (2.10).

When the mass of the adjoint Majorana fermion is zero, we conjecture the theory flows to a quantum phase that consists of three sectors:

- A Majorana fermion $\bar{\chi}$ which is the Goldstino for spontaneously broken $\mathcal{N} = 1$ supersymmetry [30].

- A TQFT which is $SO(3)_3 = SU(2)_6/\mathbb{Z}_2$ Chern-Simons theory.

- A real compact scalar $\varphi \sim \varphi + 2\pi$ denoted below by the $S^1$ sigma model.

Actually, the real scalar $\varphi$ and the TQFT $SO(3)_3$ are coupled and the precise description of the phase is the Goldstino and:

$$\frac{SU(2)_6 \times S^1}{\mathbb{Z}_2} = U(2)_{6,0} \,. \tag{4.1}$$

Above, the $\mathbb{Z}_2$ quotient means gauging the diagonal $\mathbb{Z}_2$ one-form symmetry of $SU(2)_6$ and the $U(1)$ one-form symmetry of the real scalar theory generated by the current $\star d\varphi$.

Note that the $S^1$ sigma model is time-reversal invariant, and $SO(3)_3$ is also time-reversal invariant by level-rank duality [7]. Therefore, the theory (4.1) is also time-reversal invariant as expected from the time-reversal symmetry of the UV theory. We return to this point below.

As we move away from the point $m = 0$, supersymmetry is broken and the Goldstino acquires a mass leaving only (4.1) as dynamical degrees of freedom at long distances. As $|m|$ is increased the quantum phase ends at two phase transition points that have descriptions as $U(2)$ Chern-Simons theory with one fundamental fermion. The dual descriptions have non-anomalous $U(1)$ magnetic symmetry, which is an emergent symmetry in the $G_2$ theory. This symmetry is spontaneously broken in the quantum phase with the free scalar the corresponding Nambu-Goldstone boson. Finally, after passing through these phases transitions we arrive at $U(2)$ topological field theories which are dual by (2.10) to the expected $(G_2)_{\pm 2}$ visible semiclassically. Figure 1 depicts the proposed phase structure.

There are several consistency checks on this phase diagram:

- Time-reversal symmetry and anomaly. The theory at $m = 0$ is $\mathsf{T}$ invariant and therefore has a time-reversal anomaly $\nu$ which is an integer modulo 16 [58–62]. This anomaly must match between the UV and IR. In the UV we find $\nu_{UV} = 14$ (the number of UV fermions). Meanwhile, the proposed IR phase is $\mathsf{T}$ invariant by level-rank duality $SO(3)_3 \leftrightarrow SO(3)_{-3}$, and moreover this theory has $\nu = \pm 3$ [63]. Combining with the $\nu = 1$ theory of the free Goldstino $\chi$ we find $\nu_{IR} = -2$ which agrees with $\nu_{UV}$ modulo 16.

- Gravitational Chern-Simons counterterm. In the UV it is clear that the theory at large positive $m$ and large negative mass differ by 14 units of gravitational Chern-Simons counterterm: $14\,\mathrm{CS}_{\mathrm{grav}}$. This difference by the counterterm should be reproduced by

the proposed IR phase diagram. If we set the amount of the counterterm in the UV theory to be zero as a reference, we get $7\,\mathrm{CS_{grav}}$ when $m \to \infty$, and $-7\,\mathrm{CS_{grav}}$ when $m \to -\infty$. The amount of the counterterm in other phases and other duality frames are also indicated in Figure 1. When the mass crosses each quantum transition point $U(2)_{\pm\frac{13}{2},\pm\frac{1}{2}} + \Psi_{\mathbf{fund}}$, the theory gets an additional $4\,\mathrm{CS_{grav}}$. While, in the middle phase, we use the time-reversal level-rank duality

$$U(2)_{6,0} \longleftrightarrow U(2)_{-6,0} - 9\,\mathrm{CS_{grav}} \tag{4.2}$$

discussed above to track the coefficient of $\mathrm{CS_{grav}}$. One can thus verify that the gravitational counterterm works out consistently from Figure 1.[17]

## 4.2 $(G_2)_0$ with two fundamental fermions

Let us also study the $(G_2)_0$ gauge theory with two Majorana fermions $\psi_7^i$ in the fundamental representation $\mathbf{7}$ of $G_2$. This theory has two mass parameters $m_1$ and $m_2$ for each fermion, and a $U(1)$ global symmetry rotating the two fermions when $m_1 = m_2$. We set the amount of the gravitational counterterm in the UV to be zero as a reference.

We first consider the case where the masses are restricted to $m_1 = m_2$, and subsequently generalize. We find that for small mass there is a quantum phase where the $U(1)$ global symmetry is spontaneously broken. This is similar to the analysis for classical groups in [31].

When the masses are large and positive $m_1 = m_2 \to \infty$, we have the TQFT $(G_2)_1$ TQFT, and when $m_1 = m_2 \to -\infty$, we find the TQFT $(G_2)_{-1}$. We can use the duality (2.10) to go to the $U(2)_{\pm3,\pm1}$ description. We propose that between these two phases there is a quantum phase described by the $U(2)_{2,0} = \frac{SU(2)_2 \times S^1}{\mathbb{Z}_2}$ theory. The $S^1$ valued scalar field should be identified with the Goldstone boson for the $U(1)$ symmetry.[18] From the semi-classical phase to the quantum phase, we expect a transition described by $U(2)_{\mp\frac{5}{2},\mp\frac{1}{2}}$ with a complex fermion $\Psi_{\mathbf{fund}}$ in the fundamental representation. The $U(1)$ global symmetry in UV should be identified with the monopole symmetry in the $U(2)$ description. The phase structure is summarized in Figure 2.

As in the previous subsection, we perform the following consistency checks:

- Time-reversal symmetry: The UV theory at $m_1 = m_2 = 0$ is $\mathsf{T}$ invariant. It also enjoys a discrete $\mathbb{Z}_2$ symmetry $C$ which exchanges the two fermions. The time-reversal symmetry $C\mathsf{T}$ of the UV theory is preserved provided $m_1 = -m_2$ and acts trivially on the system when $|m_i| \to \infty$ (this is because there the system itself is trivial in the IR, see below).[19] Therefore we conclude that $C\mathsf{T}$ has vanishing $\nu$.

  In the IR, the proposed phase $U(2)_{2,0} = \frac{SU(2)_2 \times S^1}{\mathbb{Z}_2}$ is also time-reversal invariant since both the $S^1$ scalar and $\frac{SU(2)_2}{\mathbb{Z}_2} = SO(3)_1$ TQFT, (which is actually a trivial TQFT), are time-reversal invariant. We refer to this time-reversal symmetry as $\mathsf{T}_{IR}$; it has $\nu = 0$. It is thus consistent to expect that the UV symmetry $C\mathsf{T}$ flows at long distances to the IR symmetry $\mathsf{T}_{IR}$.[20]

- Gravitational Chern-Simons counterterm: From the UV description, the jump of the counterterm when the mass goes from $m_1 = m_2 \to -\infty$ to $m_1 = m_2 \to \infty$ is $14\,\mathrm{CS_{grav}}$.

---

[17]To verify this, we assume that the mass of the Goldstino $\tilde{\chi}$ is negatively related to the UV mass $m$.

[18]For this analysis that follows to be correct, it is important that the $\mathbb{Z}_2$ subgroup of the UV $U(1)$ symmetry is not spontaneously broken in the quantum phase. Instead, we identify this unbroken $\mathbb{Z}_2$ 0-form symmetry of the UV with the $\mathbb{Z}_2$ 0-form symmetry that arises from the $\mathbb{Z}_2$ 1-form gauging in the IR, which defines the quotient $\frac{SU(2)_2 \times S^1}{\mathbb{Z}_2}$.

[19]The symmetry $\mathsf{T}$ maps $m_i \to -m_i$ while $C$ exchanges the $m_i$. Hence $C\mathsf{T}$ is a symmetry if $m_1 = -m_2$.

[20]The UV theory also has the anomalous $\mathsf{T}$ symmetry with $\nu=14$. However it is possible that this is matched in the IR by mixing with the 1-form symmetry of the massless scalar (see e.g. [34, 64, 65]).

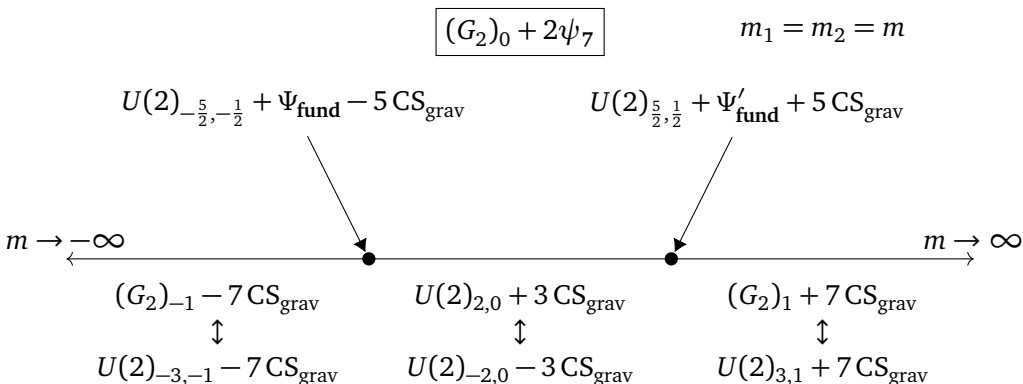

Figure 2: A conjectural phase diagram of $(G_2)_0$ gauge theory with 2 fermions in the fundamental representation where $m_1 = m_2$. In the center is a quantum phase where the $U(1)$ global symmetry is spontaneously broken.

One can verify that the same amount of counterterm is generated when the system goes through the two transition points into the quantum phase from Figure 2.

Next we would like to generalize the phase diagram into where $m_1 \neq m_2$. Assuming Figure 2 we further deduce:

- If we set $m_1 = -m_2$ and take the limit of $m_1 = -m_2 \to \pm\infty$, in the theory becomes $(G_2)_0$ Yang-Mills theory. In particular it is confined and trivially gapped in the IR and the coefficient of the gravitational Chern-Simons term is zero there.

- If we add a small $U(1)$ breaking mass term $m_1 - m_2$ to the quantum phase $U(2)_{2,0} + 3\,\mathrm{CS}_{\mathrm{grav}}$, the $S^1$ valued Goldstone scalar gets mass, and we are left with $\frac{SU(2)_2}{\mathbb{Z}_2} + 3\,\mathrm{CS}_{\mathrm{grav}} = SO(3)_1 + 3\,\mathrm{CS}_{\mathrm{grav}} = 0\,\mathrm{CS}_{\mathrm{grav}}$, namely the trivial theory with zero amount of gravitational Chern-Simons counterterm.

Therefore, it is consistent to smoothly connect the phase $m_1 = -m_2 = \pm\infty$ and the phase with small $m_1 = -m_2$.

Thus we propose the phase diagram for $(G_2)_0 + 2\psi_7$ with general masses as Figure 3. On the $m_1 \to \pm\infty$ line, we reduce to $(G_2)_{\pm\frac{1}{2}} + \psi_7$. On the line, we expect a transition point where the fermion mass changes sign, between the semiclassical $(G_2)_{\pm 1}$ and $(G_2)_0$ phases. If we assume that this transition point continues to exist when $|m_2|$ is reduced until we intersect the line of $U(1)$ global symmetry, we get the diagram Figure 3. If this diagram is correct, the monopole operator of the theory $U(2)_{\pm\frac{5}{2},\pm\frac{1}{2}} + \Psi_{\mathbf{fund}}$ should be relevant, and it triggers a flow to $(G_2)_{\pm\frac{1}{2}} + \psi_7$ with a tuned mass for $\Psi_{\mathbf{fund}}$.

## Acknowledgements

We thank N. Seiberg for discussions. C.C. is supported by DOE grant de-sc0009988. P.-S.H. is supported by the U.S. Department of Energy, Office of Science, Office of High Energy Physics, under Award Number DE-SC0011632, and by the Simons Foundation through the Simons Investigator Award. K.O. is supported by NSF Grant PHY-1606531 and the Paul Dirac fund.

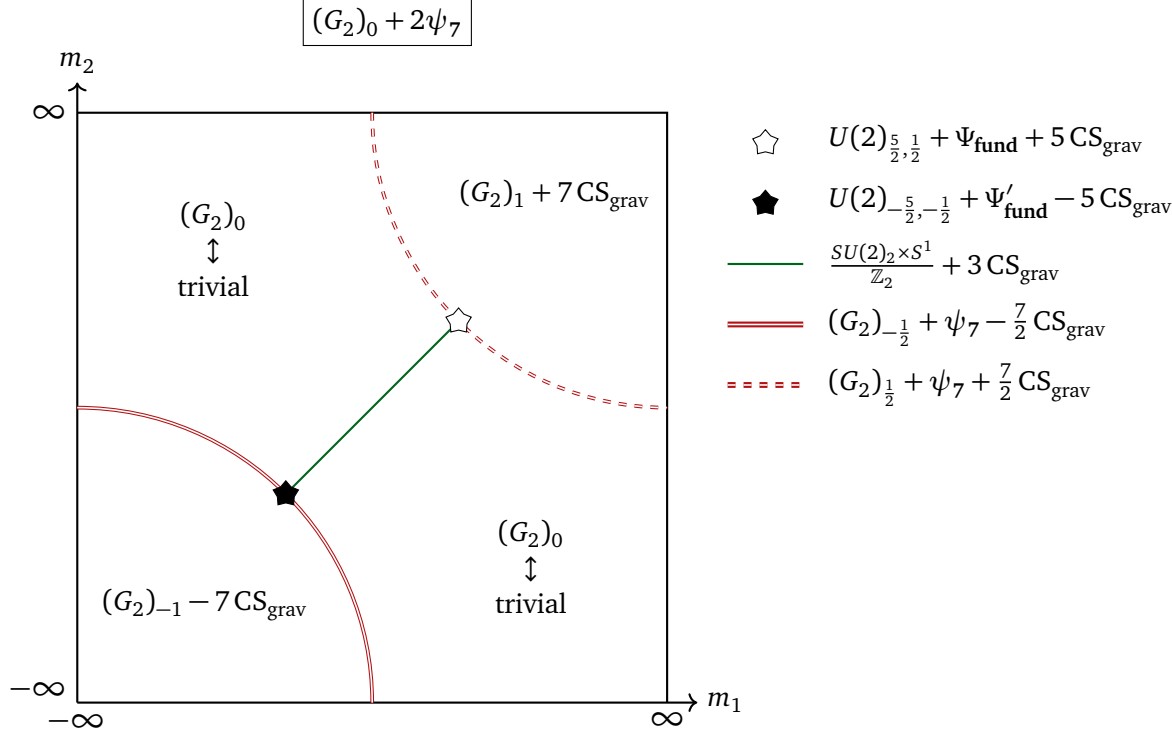

Figure 3: A conjectural phase diagram for the $(G_2)_0$ gauge theory with two real fermions $2\psi_7$ in the fundamental representation of $G_2$. The diagonal line where $m_1 = m_2$ is identical to Figure 2. As explained there, on the solid line in the middle, the theory flows to $\frac{SU(2)_2 \times S^1}{\mathbb{Z}_2}$, and at the black and white stars the theory is dual to $U(2)_{\pm\frac{5}{2},\pm\frac{1}{2}} + \Psi_{\mathbf{fund}}$. On the solid and dashed doubled lines we expect that the theory flows to a gapless theory dual to $(G_2)_{\pm\frac{1}{2}} + \psi_7$, and intersects with the $m_1 = m_2$ line at the black and white stars.

## A   Potential for Higgsing

Consider a real scalar field $\phi$ taking value in an orthogonal representation $R$ of a compact gauge group $G$. In the main text we often need to find a one-parameter family of potentials $V(\phi; m^2)$ such that with $m^2 \to +\infty$ the scalar is massive and decouples, and with $m^2 \to -\infty$ the gauge group $G$ is Higgsed down to a subgroup $H$. It is necessary for such a $V$ to exist that the representation $R$ includes a vector whose stabilizer is $H$. Below we show that actually the converse is also true.

Let $v$ be a vector in $R$ with stabilizer $H$ and norm $|v| = 1$. Denote the $G$-orbit of $v$ by $\mathcal{O}(v)$. The potential $V(\phi; m^2)$ with the desired property can be constructed as

$$V(\phi; m^2) = \begin{cases} \max_{y \in \mathcal{O}(v) \cup \mathcal{O}(-v)} (|\phi + m^2 y|^2)^2 & m^2 > 0 \\ |\phi|^4 & m^2 = 0 \\ \min_{y \in \mathcal{O}(v) \cup \mathcal{O}(-v)} (|\phi + m^2 y|^2)^2 & m^2 < 0 \,. \end{cases} \tag{A.1}$$

When $m^2 > 0$, for any vector $\phi$ and $y$ either $|\phi + m^2 y|^2$ or $|\phi - m^2 y|^2$ is greater than $|m^2 y|^2 (= m^4)$. Therefore, $\phi = 0$ is the minimum point of $V(\phi, m^2 > 0)$. Moreover, as $m^2$ goes to infinity, the mass of $\phi$ also goes to infinity. On the other hand, when $m^2 < 0$, $V$ has minimum 0 if and only if $\phi/m^2 \in \mathcal{O}(v) \cup \mathcal{O}(-v)$. Any element of $\mathcal{O}(v) \cup \mathcal{O}(-v)$ has stabilizer $H$, so classically the gauge group is Higgsed down to $H$. Moreover, when $m^2 \ll 0$, $\phi$ gets a

large vev, and therefore the classical analysis is reliable. The potential $V(\phi; m^2)$ constructed above is not necessarily smooth, but it should be possible find a nearby smooth potential if needed.

When $m^2$ is close to zero, we cannot say anything concrete about the quantum theory. The purpose of this appendix is to explicitly show that the semiclassical phase with $G$-TQFT and $H$-TQFT are continuously connected by the potential deformation, and therefore we expect some phase transition connecting them.

For a complex scalar $\Phi$ valued in a unitary representation, a similar $V$ can be written as

$$V(\Phi; m^2) = \begin{cases} \max_{y \in \cup_\alpha \mathcal{O}(e^{i\alpha}v)} (|\Phi + m^2 y|^2)^2 & m^2 > 0 \\ |\Phi|^4 & m^2 = 0 \\ \min_{y \in \cup_\alpha \mathcal{O}(e^{i\alpha}v)} (|\Phi + m^2 y|^2)^2 & m^2 < 0 \,. \end{cases} \tag{A.2}$$

This potential preserves the $U(1)$ symmetry rotating $\Phi$ to $e^{i\alpha}\Phi$.

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
