# Peer review of "Exceptional Chern-Simons-Matter Dualities"

_SciPost Physics, doi:SciPost Phys. 7, 056 (2019)_

## Round 1 · Referee Report · Anonymous (Referee 1) · 2019-8-30

Strengths

  1. The paper reports on the new results, potentially interesting for further developments in the field.
  2. It contains a review of some underlying material, which facilitates the presentation.

Weaknesses

  1. At some points the argument flow is not very detailed.

Report

In the paper "Exceptional Chern-Simons-Matter Dualities" the authors start by adding to the previously known families of dualities between Chern-Simons theories a set of new ones, involving in particular Chern-Simons theories with exceptional gauge groups. This is done exploiting the conformal embedding technique for the corresponding chiral algebras, as well as with the aid of some results concerning more formal aspects of Kac-Moody algebras. Armed with the new dualities for topological Chern-Simons theories, the authors proceed with formulating an array of conjectures regarding the Chern-Simons-matter dualities with scalar matter. Finally, they formulate conjectures regarding the phase spaces of the $(G_2)_0$ theory coupled to an adjoint fermion and of $(G_2)_0$ theory coupled to two fundamental fermions.
I didn't find any points requiring significant revision, but I would like to indicate a few possible typos or omissions.
Page 8, Section 2.3, the last line of the second paragraph. The index $r$ must run from $0$ to $8-N$.
Probably the duality $Spin(10)_1 \longleftrightarrow SU(4)_1$ discussed around eq. 2.15 should be included in the lists of dualities 1.3 and 2.5.
Page 22, Ref. [9] has an excessive list of authors.

The above mentioned points don't lower quality of the work. The paper is well-written and contain a lot of important results, which will serve as the ground for further developments in the subject.  I therefore recommend the paper for publishing.

Requested changes

See the report.

---

## Round 1 · Referee Report · Anonymous (Referee 2) · 2019-9-2

Strengths

1- Fills in an conspicuous gap in the literature - regarding how the exceptional groups fit into the picture of TQFTs for the classical Lie groups - in a satisfying way. 2- The paper takes the time to briefly recapitulate the basic story of dualities of chiral algebras, and also spells out the mapping of lines across some of the dualities, both of which will be helpful in making the paper more useful to the community.

Weaknesses

1- By the nature of the subject, the core content of the paper is conjectural, but this is mostly not a very serious criticism, as the conjectures are reasonable, the available evidence is encouraging, and these conjectures are helpful in directing future research. 2- One more specific weak point (see also the first requested change) is the lack of any discussion of other classically marginal operators.

Report

This is a nice, well-motivated paper seeking to address some of the remaining questions of what has been a fashionable subject in recent years, namely Chern-Simons-matter dualities. The authors first give a concise, helpful presentation of some mathematically rigorous facts about topological quantum field theories with exceptional gauge groups. They then use this (in the spirit of various papers from the last couple of years) to conjecture dualities between matter theories which, in the IR, interpolate between those TQFTs as parameters are varied; and also to explore a phase diagram of some fermionic theories. The authors have presented the material well and justify their conjectures to a similar standard to what is currently expected in the field, though a little more consideration of classically marginal operators (see other parts of the report) may help clarify this.

Requested changes

1- It is stated that "quartic" couplings ought to be sufficient to achieve certain symmetry breaking patterns, and Appendix A is offered to justify this. But it is not clear that (A.1) is always a quartic function, and it is commented that "The potential V constructed above is not necessarily smooth". It would be good to clarify these comments. 2- A natural related question is whether one might have any different quartic operators being relevant and hence needing tuning to observe the physics proposed here. Any comment on this would be a nice bonus. 3- It might be nice to highlight other ways one might seek evidence for these conjectures. 4- Page 8: "antisymemtric" is a typo. 5- Throughout, "Higgs"/"higgs" and "Goldstino"/"goldstino" should probably have consistent capitalization.

---

## Round 2 · List of Changes

- new footnote 8
- replace the word quartic coupling by a suitable potential
- fixed typos mentioned in the report
- update references

You are currently on this page

Resubmission 1812.11705v2 on 19 September 2019

---

## Editorial Decision

published